# Apoptotic Cells induce Proliferation of Peritoneal Macrophages

**DOI:** 10.3390/ijms22052230

**Published:** 2021-02-24

**Authors:** Anne-Kathrin Knuth, Arnaud Huard, Zumer Naeem, Peter Rappl, Rebekka Bauer, Ana Carolina Mota, Tobias Schmid, Ingrid Fleming, Bernhard Brüne, Simone Fulda, Andreas Weigert

**Affiliations:** 1Institute for Experimental Cancer Research in Pediatrics, Goethe-University, Komturstraße 3a, 60528 Frankfurt, Germany; a.knuth@kinderkrebsstiftung-frankfurt.de; 2Faculty of Medicine, Institute of Biochemistry I, Goethe-University, Theodor-Stern-Kai 7, 60590 Frankfurt, Germany; huard@biochem.uni-frankfurt.de (A.H.); rappl@biochem.uni-frankfurt.de (P.R.); bauer@biochem.uni-frankfurt.de (R.B.); ac.mota92@gmail.com (A.C.M.); t.schmid@biochem.uni-frankfurt.de (T.S.); b.bruene@biochem.uni-frankfurt.de (B.B.); 3Institute for Vascular Signaling, Department of Molecular Medicine, Goethe-University, Theodor-Stern-Kai 7, 60590 Frankfurt, Germany; Naeem@vrc.uni-frankfurt.de (Z.N.); Fleming@vrc.uni-frankfurt.de (I.F.)

**Keywords:** apoptosis, peritoneal macrophages, RNA sequencing, Zymosan-induced peritonitis, proliferation

## Abstract

The interaction of macrophages with apoptotic cells is required for efficient resolution of inflammation. While apoptotic cell removal prevents inflammation due to secondary necrosis, it also alters the macrophage phenotype to hinder further inflammatory reactions. The interaction between apoptotic cells and macrophages is often studied by chemical or biological induction of apoptosis, which may introduce artifacts by affecting the macrophages as well and/or triggering unrelated signaling pathways. Here, we set up a pure cell death system in which NIH 3T3 cells expressing dimerizable Caspase-8 were co-cultured with peritoneal macrophages in a transwell system. Phenotype changes in macrophages induced by apoptotic cells were evaluated by RNA sequencing, which revealed an unexpectedly dominant impact on macrophage proliferation. This was confirmed in functional assays with primary peritoneal macrophages and IC-21 macrophages. Moreover, inhibition of apoptosis during Zymosan-induced peritonitis in mice decreased mRNA levels of cell cycle mediators in peritoneal macrophages. Proliferation of macrophages in response to apoptotic cells may be important to increase macrophage numbers in order to allow efficient clearance and resolution of inflammation.

## 1. Introduction

Inflammation is a regulated reaction to eliminate infection and to combat other disturbances of tissue homeostasis. A successful resolution of inflammation is necessary to restore homeostasis and prevent the development of chronic inflammation or autoimmune disease. During inflammation, polymorphonuclear leukocytes (PMNs) infiltrate the affected tissue to phagocytose and kill invading pathogens. Migration of neutrophils and monocytes into tissues is directed through the release of chemokines by cells in the inflamed tissue [1]. To end inflammation, neutrophils undergo apoptosis and are removed by tissue-resident or monocyte-derived macrophages via efferocytosis. This step is important for the resolution of inflammation because it prevents the release of danger associated molecular patterns (DAMPs) by secondary necrotic cells and avoids ongoing secretion of pro-inflammatory cytokines [1,2].

Macrophages are important cells of the immune system that, in addition to their role in the phagocytosis of pathogens, debris or dead cells, also present antigens and produce cytokines and chemokines [3]. Macrophages react dynamically to stimuli in their environment, upon which they change their gene expression profiles and metabolism in a process called macrophage polarization. In a simplified classification system, macrophages can be divided into classically activated M1 macrophages and alternatively activated M2 macrophages. Tumor necrosis factor (TNF) α or lipopolysaccharide (LPS) in combination with interferon (IFN) γ lead to a classical activation of macrophages. These cells protect against pathogens by phagocytosis and antigen presentation. M1 macrophages have a pro-inflammatory phenotype and express high levels of pro-inflammatory cytokines such as TNF-α, interleukin (IL) 1β and IL-6. Efficient efferocytosis leads to changes in the polarization of macrophages from M1 towards a pro-resolving M2-like phenotype. M2 macrophages are usually activated by IL-4, IL-10, or IL-13 and participate in debris clearance and long-term tissue repair by decreased antigen-presentation and production of pro-resolving cytokines, e.g., IL-10, transforming growth factor (TGF) β, and specialized pro-resolving lipid mediators [3,4,5]. Apoptotic cells mediate reduced nitric oxide (NO) production in macrophages by decreasing the concentration of NO synthase substrate L-arginine as a consequence of arginase expression. They also inhibit reactive oxygen species (ROS) production through a peroxisome proliferator-activated receptor (PPAR) γ-dependent mechanism and suppress the expression of pro-inflammatory cytokines [6]. Apoptotic cells thus repress inflammatory responses in macrophages. However, the molecular signaling events controlling these processes require further analysis.

In addition to polarization, macrophage numbers are also dynamically regulated during inflammation. An increase in macrophage numbers can result from the influx of blood borne monocytes or the proliferation of tissue-resident macrophages. Extrinsic factors such as macrophage colony-stimulating factor (M-CSF), granulocyte-macrophage colony-stimulating factor (GM-CSF), and IL-4, and intrinsic factors, including the activation of certain transcription factors are involved in the induction of macrophage proliferation [7].

Apoptosis is a regulated form of cell death that is usually anti-inflammatory and depends on the activation of caspases. Intrinsic apoptosis is induced upon the detection of intracellular damage signals, whereas extrinsic apoptosis results from recognition of danger by the immune system and the subsequent ligation of death receptors. After detection of danger or damage, initiator caspases (8 or 9) are activated [8]. The aspartate-specific cysteine protease caspase-8 is expressed as inactive zymogen, consisting of an N-terminal pro-domain with two death effector domains (DEDs), a large protease subunit and a small subunit. To be activated caspase-8 needs to bind to the death-inducing signaling complex (DISC), consisting of Fas-associated death domain protein (FADD) and a death receptor, whereupon it forms homodimers and undergoes autocatalytic cleavage between the large and small subunits [9]. The initiator caspases activate executioner caspases 3, 6 and 7 by protein cleavage, which leads to DNA fragmentation, destruction of proteins, phosphatidylserine externalization and formation of apoptotic bodies [8]. Phosphatidylserine on the surface of apoptotic cells functions as eat me signal for macrophages. In vivo, macrophages phagocyte apoptotic cells before they are fully fragmented to avoid disintegration [5].

During development and homeostasis, caspase activation induces proliferation by induction of mitogenic signals to reinsure tissue regeneration. In this manner, caspases can stimulate the proliferation of the cells themselves or of cells in the direct neighborhood. Apoptosis also induces proliferation of lymphocytes; however, the underlying signals and mechanisms remain to be elucidated [10,11].

Most of the past studies used activators of death receptors, e.g., Fas or TNF-α, or chemical agents to induce apoptosis and investigate the effect of apoptotic cells on immune cells. However, such approaches have the inherent disadvantage that several other signaling pathways were also activated [12]. Here, we used a system, in which apoptosis was selectively induced by the dimerization of caspase-8 [13], in combination with an unbiased approach. A co-culture model of macrophages and dimerizer-induced apoptotic cells in a transwell system was established. RNA sequencing of macrophages after co-culture with apoptotic cells revealed enhanced proliferation of macrophages. This finding was confirmed in an in vivo model of Zymosan-induced peritonitis at molecular level, where macrophages face dying neutrophils during the resolution of inflammation.

## 2. Results

### 2.1. Generation of a Dimerizer-Induced Apoptotic System

The murine fibroblast cell line NIH 3T3, stably expressing human Caspase-8 with the DEAD domains being replaced by a dimerizer domain (FK506 binding protein, FKBP), was generated using lentiviral transduction to create a pure apoptotic cell death system (Figure 1A). Addition of dimerizer leads to dimerization and activation of Caspase-8 and thereby induces apoptosis. Two single clones were selected based on their efficiency of protein expression of human Caspase-8 and co-expression of green fluorescent protein (GFP, Figure 1B; Appendix A). Both single clones were dying in a time dependent manner after treatment with dimerizer, showing significant apoptosis (Annexin V positive, while being PI negative) starting from 6 h after treatment, with membrane integrity loss starting after 8 h (Figure 1C,D). Efficient induction of cell death was validated by investigating executioner caspase activation (Figure 1E). Importantly, the activation of executioner caspases 3/7 and cell death could be blocked by the pan-caspase inhibitor zVAD.FMK (Figure 1E,F). Together, these data show that the dimerization of Caspase-8 induced apoptosis in NIH 3T3 cells.

### 2.2. Co-Culture with Apoptotic Cells Induces Macrophage Proliferation

Next, we established a co-culture system to study the effect of apoptotic cells on macrophages. Therefore, peritoneal macrophages were isolated from wild-type mice and co-cultured with NIH 3T3 cells in a transwell system prior to the addition of dimerizer (Figure 2A). After 48 h, macrophages released the cytokine CXCL5 and expressed elevated levels of the anti-inflammatory macrophage polarization marker CD206 (Figure 2B,C). Thus, the co-culture protocol effectively polarized the peritoneal macrophages as the release of CXCL5 and induction of CD206 were previously described as macrophage response to apoptotic cells [5,14].

To gain new insights into gene expression profiles of macrophages after contact with apoptotic cells, macrophages were either treated with dimerizer alone or co-cultured with apoptotic cells and the macrophage transcriptome was analyzed using whole genome mRNA sequencing (RNA-Seq; Figure 3A). To assess whether the co-culture with apoptotic cells represents the major source of variation, principal component analysis (PCA) was performed using normalized RNA-Seq data of differentially expressed genes (Figure 3B). Sample groups showed a clear separation and the variation between biological replicates (macrophages from individual animals) was small. MA plotting comparing fold-change values of genes against the mean of normalized counts of all samples were then applied (Figure 3C). A large number of genes showed a significant change in gene expression (2410 upregulated, 2359 downregulated; Appendix A). The reproducibility of biological replicates was again validated by hierarchical clustering (Figure 3D). Replicates of control and treatment groups clustered together. These analyses confirmed that the co-culture with apoptotic cells lead to profound changes in gene expression in peritoneal macrophages.

To analyze the pathways underlying the altered gene pattern, Gene set enrichment analyses (GSEA) and Gene ontology (GO) analyses were performed (Figure 4). GSEA showed enrichment of the prostaglandin E2 pathway (Figure 4A, Appendix A), which is known to be activated following efferocytosis [15]. Interestingly, cell cycle and DNA replication pathways were also enriched (Figure 4B–D). Consistently, GO analysis showed that mainly pathways related to cell cycle, cell division, and mitosis were upregulated in macrophages co-cultured with apoptotic cells (Figure 4E). Again, enrichment of gene sets suggesting apoptotic cell clearance proved the accuracy of the experimental system.

Macrophage proliferation was a neglected phenomenon for a long time, but it has recently been rediscovered. Therefore, we were interested to look deeper into the activation of cell cycle pathways. Fourteen genes associated with the cell cycle were upregulated in our study (Figure 5A). A selection of these mediators of the cell cycle were also significantly upregulated at mRNA level when RNA-Seq data was independently validated by RT-qPCR (Figure 5B). We next asked whether macrophages were indeed proliferating in the co-culture system. To explore this, confluency was analyzed using an incubator-based imaging system. Generally, co-culture with NIH 3T3 cells appeared to provide a survival benefit for peritoneal macrophages (Figure 5C). However, wells containing macrophages co-cultured with apoptotic cells reached a higher degree of confluency (an index of proliferation) than wells with macrophages co-cultured with fibroblasts, implying that the co-culture with apoptotic cells further stimulated macrophage proliferation. Furthermore, the IC-21 peritoneal macrophage cell line was stained with eFluor 670 before the co-culture and fluorescence intensity was measured after 48 h (Figure 5D, E). The fluorescent dye eFluor 670 binds to any cellular protein containing primary amines and it is diluted during cell division. Co-culture of IC-21 cells with apoptotic cells significantly decreased the mean fluorescence intensity of the dye, suggesting an accelerated proliferation of IC-21 cells (Figure 5D,E), although it is noteworthy that apparently only a small proportion of the cells actually proliferated. Together, these data indicate that the interaction with apoptotic cells induces the expression of cell cycle-related genes and proliferation of peritoneal macrophages.

### 2.3. Inhibition of Apoptosis in a Model of Self-Resolving Inflammation Reduces mRNA Expression of Cell Cycle Genes in Macrophages In Vivo

To explore the role of apoptosis in resolution of inflammation in vivo, a model of self-resolving peritonitis was applied by injecting 10 mg/kg Zymosan-A into the peritoneal cavity of C57 BL/6 mice. During the time course of inflammation in this model, the yeast component Zymosan-A is recognized by dectin-1 and toll-like receptor 2 (TLR2) on phagocytes. Activation of these receptors leads to phagocytosis, production of ROS, nuclear factor κ-B activation and, in the end, release of pro-inflammatory cytokines and chemokines [16]. The influx of neutrophils peaks within 4 h and resolution starts after 72 h. At day six, inflammation is usually resolved [17].

Eight hours after Zymosan-A injection, i.e., shortly after the peak of neutrophil influx, apoptosis was inhibited by injection of zVAD.FMK *i.p.* (Figure 6A). Cells and mediators in the peritoneum were isolated 1, 3 and 6 days after Zymosan administration. The neutrophil peak at day one indicated that inflammation had been successfully induced. Neutrophil levels decreased by day three, while percentage of macrophages, dendritic cells, B cells and T cells increased at day three accordingly, demonstrating, that resolution of inflammation started (Figure 6B, Appendix A). Injection of zVAD.FMK led to decreased levels of dead neutrophils at day one and three (Figure 6C). Next, we studied the effect of apoptosis inhibition on expression of cell cycle genes in macrophages. Inhibition of apoptosis decreased the expression of Aurora A kinase at day one, while cyclin b1 expression was decreased on days one and three. Cyclin-dependent kinase 1 (CDK1) expression was also significantly reduced on day three after Zymosan application (Figure 6D–F). In vitro, co-culture with apoptotic cells led to increased levels of CXCL5. In vivo, inhibition of apoptosis reduced levels of CXCL5 in the peritoneum on day one. In summary, these data show that inhibition of apoptosis in vivo effects markers of macrophage proliferation at the site of inflammation, suggesting that macrophage proliferation is indeed a feature of the interaction with apoptotic cells.

## 3. Discussion

In the present study, we describe a direct and prominent impact of apoptotic cells on peritoneal macrophage proliferation. This conclusion is underscored by RNA sequencing, which was validated using two different proliferation assays in peritoneal macrophages and upregulation of cell cycle mediators at mRNA level. Furthermore, inhibition of apoptosis in an in vivo model of self-resolving inflammation revealed reduced mRNA levels of cell cycle mediators in macrophages. Inhibition of apoptosis during Zymosan-induced peritonitis decreased the levels of cell cycle mediators in peritoneal macrophages but did not affect neutrophil levels or the timeline of inflammation. Proliferation of macrophages in this setting may only slightly contribute to macrophage levels as a massive influx of monocytes occurs during inflammation.

To exclude activation of other signaling pathways except apoptosis, a pure cell death system was used by addition of dimerizer to cells expressing Caspase-8 fused to a dimerizer domain. It was shown before that dimerization of Caspase-8 is sufficient to induce apoptosis. As expected, phagocytosis of these cells did not induce an inflammatory phenotype in dendritic cells [13]. We could confirm this finding in macrophages co-cultured with apoptotic cells, which did not develop a pro-inflammatory phenotype either, although cells underwent secondary necrosis at later time points. Our established co-culture system is thereby an easy-to-handle method to study the interaction of two different cell types because no complex separation process is needed to analyze mRNA levels of macrophages. The pore size of the transwell membrane was 5 μm, allowing medium, apoptotic bodies and mediators to diffuse through the membrane [18]. Co-culture of macrophages with living fibroblasts maintained their survival indicating that fibroblasts produce mediators that are beneficial for macrophages in cell culture. Apoptotic cells in the co-culture with macrophages induced higher levels of proliferation either through phagocytosis of apoptotic bodies or released mediators. Our data indicate a major impact of PGE_2_ signaling in the macrophage phenotype in response to apoptotic cells. PGE_2_ has been suggested a one driver of apoptotic cell-induced, and particularly caspase-induced proliferation [19]. Thus, PGE_2_ signaling may be a major candidate for promoting macrophage proliferation in response to apoptotic cells. However, PGE_2_ may promote or inhibit macrophage proliferation in a context-, polarization- and concentration-dependent manner [20,21,22]. Functional studies into molecular drivers of macrophage proliferation therefore will require extensive future studies.

Despite a, historically, widely held belief, that macrophages do not proliferate, our data together with other publications support proliferation of macrophages in tissues. Recent lineage tracing studies have shown that macrophages from embryonic sources sustain their numbers in tissues independently from infiltrating monocytes, thus undergoing proliferation [23]. This was described under physiological conditions, during inflammation, infection and in diseases like atherosclerosis, obesity, and cancer [24,25]. M-CSF was shown to promote macrophage proliferation in the peritoneal cavity at steady state and during Zymosan-induced peritonitis [26]. However, a potential impact of apoptosis and apoptotic cells was not studied. In the context of IL-4-inducing helminth infection, genetic ablation of phagocytic receptors, MerTK and AXL, leads to reduced proliferation of macrophages and induction of anti-inflammatory and tissue repair genes. These experiments indicate that phagocytosis of apoptotic cells is necessary to induce tissue repair programs in macrophages at least in the presence of IL-4 [27], supporting findings in our system that efferocytosis can lead to induction of macrophage proliferation. Moreover, phagocytosis of microbes stimulated macrophages to re-enter the cell cycle, hypothesizing a phagocytosis-induced activation of proliferation, although the mechanisms of bacteria phagocytosis and efferocytosis are fundamentally different [28]. Activation of macrophage proliferation may also be context- and tissue-specific as Reddy and colleagues described that phagocytosis of apoptotic cells inhibits cell cycle entry in macrophages [28]. Our data are not fully comparable with this publication, as they used peritoneal macrophages isolated after thioglycolate challenge and induced apoptosis by irradiation or staurosporine treatment, which are not pure cell death systems.

The reason why apoptotic cells would induce proliferation in macrophages is unclear. This may be a bystander effect of caspase-induced proliferation that was described previously during development and tissue homeostasis [10,11]. Alternatively, an increase in macrophage numbers may help to remove dead cells and supports tissue regeneration and resolution of inflammation. Moreover, self-renewal could be important to ensure an efficient local macrophage population allowing quick immune responses to secondary infections. However, increased numbers of anti-inflammatory cells such as macrophages in the anti-inflammatory phase of inflammation can also have detrimental consequences. In atherosclerosis it was shown that macrophage accumulation in arterial walls is not only depending on recruitment of monocytes but also on proliferation of macrophages [29]. Thus, macrophage proliferation can lead to chronic inflammation. Furthermore, fibrosis can be the result of an incomplete resolution of inflammation, which can be induced by excess numbers of macrophages [30,31]. Additionally, in adipose tissue excess macrophages can induce inflammation [24]. In cancer, proliferation of resident macrophages was shown to ensure maintenance of tumor-associated macrophages (TAMs). These TAMs are also subjected to large numbers of apoptotic tumor cells, which can induce a pro-tumor phenotype and promote tumor growth [32]. The role of cell death or apoptosis in inducing macrophage proliferation in these diseases therefore needs to be studied further.

## 4. Materials and Methods

### 4.1. Cell Culture and Chemicals

NIH 3T3 cells were obtained from DSMZ (Deutsche Sammlung von Mikroorganismen und Zellkulturen, Braunschweig, Germany). IC-21 and Phoenix AMPHO cells were obtained from ATCC (American Type Culture Collection, CEM, Manassas, VA, USA). NIH 3T3 and Phoenix AMPHO cells were cultured in DMEM Medium (Life Technologies, Inc., Eggenstein, Germany) supplemented with 10% fetal calf serum (FCS, Life Technologies), 1% sodium pyruvate (Life Technologies) and 1% penicillin/streptavidin (Life Technologies). Peritoneal macrophages, isolated from C57 BL/6 mice using 3 or 5 mL PBS, and IC-21 cells were cultured in RPMI Medium 1640 GlutaMAXTM-I (Life Technologies) supplemented with 10% FCS and 1% penicillin/streptavidin. Cell lines were negatively tested for mycoplasma contamination. zVAD.FMK was obtained from Bachem (Heidelberg, Germany), B/B Homodimerizer from Clontech (Mountain View, CA, USA), Annexin V-FITC from Immunotools (Friesoythe, Germany) and Annexin V-APC from Biolegend (San Diego, CA, USA). All other chemicals were obtained from Sigma-Aldrich (Taufkirchen, Germany) or Carl Roth (Karlsruhe, Germany), unless otherwise indicated.

### 4.2. Generation of NIH 3T3 Cells Expressing FV-hCasp8

Plasmids to generate NIH 3T3 FV-hCasp8 cells (FV-hCasp8-2A-GFP in pBabe Puro) were kindly provided by Andrew Oberst (University of Washington, [13]). Transfection of Phoenix cells was performed using FuGENE transfection reagent (Promega, Madison, WI, USA) and 2 μg of DNA in Opti-MEM reduced media (Gibco, Carlsbad, CA, USA). Medium was exchanged after 24 h and supernatant containing virus was harvested 48 and 72 h after transfection. Supernatant was filtrated using a 0.45 μm filter. NIH 3T3 cells were transduced with supernatant containing virus using 10 μg/mL polybrene. 72 h after transduction, medium was supplemented with 1 μg/mL puromycin to select for FV-hCasp8 positive cells. After 7 days, single clones were generated by seeding 1.2 cells per well in a 96 well plate. Two different single clones were selected by analyzing expression of FV-hCasp8 and GFP protein levels and cell death by B/B Homodimerizer treatment.

### 4.3. Co-Culture of Peritoneal Macrophages with Apoptotic Cells

Cells of the peritoneum of C57BL/6 mice were isolated by lavage with 3 or 5 mL PBS and 1 × 10^6^ or 0.5 × 10^6^ cells were seeded in 24 well plates. After 1 to 4 h nonadherent cells were washed away by washing 3 times with PBS: Remaining cells were defined as peritoneal macrophages (0.5 × 10^6^ or 0.25 × 10^6^ peritoneal macrophages/well), which was routinely controlled by flow cytometry using the markers employed to identify macrophages in the peritoneum. Macrophages were co-cultured at a 1 to 1 ratio with NIH 3T3 FV-hCasp-8 cells in a transwell system (6.5 mm transwell with 5 μm pore size; Corning, Corning, NY, USA). After an overnight co-culture, 10 nM B/B Homodimerizer was added. 48 h after treatment, chemokines in the supernatant, mRNA level of macrophages and confluence of macrophages were analyzed.

### 4.4. Zymosan-A-Induced Peritonitis

Experiments involving mice were approved by and followed the guidelines of the Hessian animal care and use committee (FU1244, 22 July 2019). To induce a self-resolving inflammation in mice, 10 mg/kg Zymosan-A in 200 μL PBS were injected *i.p.* into male and female C57 BL/6 mice (8 to 12 weeks old, >18 g body weight). 8 h after Zymosan-A injection, 10 mg/mL zVAD.FMK were injected *i.p*. to inhibit apoptosis. 1, 3, or 6 days after Zymosan-A injection, chemokines and cells in the peritoneum were isolated via lavage using 3 mL ice-cold PBS. For determination of the composition of immune cells in the lavage, cells were stained with different antibodies (Table 1) and analyzed using flow cytometry (BD, LSRFortessa). Neutrophils were defined as CD45^+^Ly-6G^+^Ly-6C^+^MHC-II^-^. Data were analyzed using FlowJo version 10.6.2.

### 4.5. Western Blot Analysis

Western blot analysis was performed as described previously [33] using the following antibodies: Mouse anti-human Caspase-8 (Enzo, Farmingdale, NY, USA), rabbit anti-GFP (Clontech), and mouse anti-β-Actin (Sigma-Aldrich). Goat anti-mouse IgG or goat anti-rabbit IgG conjugated to horseradish peroxidase (Santa Cruz Biotechnologies, Dallas, TX, USA) and enhanced chemiluminescence (Amersham Bioscience, Freiburg, Germany) or infrared dye-labeled secondary antibodies and infrared imaging (Odyssey Imaging System, LI-COR Bioscience, Bad Homburg, Germany) were used for detection.

### 4.6. Cell Death Analysis

0.5 × 10^6^ NIH 3T3 FV-hCasp-8 cells/mL were seeded in 96 or 24 well plates and treated with 10 nM B/B Homodimerizer for indicated time points. Cell death was analyzed by staining with 1 μg/mL propidium iodide (PI) and co-staining of the nucleus with 10 μg/mL Hoechst and fluorescence based microscopy using ImageXpress Micro XLS Analysis System and MetaXpress Software according to the manufacturer’s instructions. Annexin V-positive cells were analyzed by co-staining of PI and Annexin V-APC or Annexin V-FITC using flow cytometry (BD, FACSCanto II or LSRFortessa). Data were analyzed using FlowJo version 10.6.2.

### 4.7. Caspase-Activity Assay

0.5 × 10^6^ NIH 3T3 FV-hCasp-8 cells/mL were stained with 2 μM Cell Event Caspase 3/7 Green Detection Reagent (Life Technologie), seeded in a 96 well plate and treated with 10 nM B/B Homodimerizer for indicated time points. At the end of the experiment, cells were stained with 1 μg/mL Hoechst and Caspase 3/7 active cells were analyzed using fluorescence based microscopy (ImageXpress Micro XLS Analysis System) and MetaXpress Software according to the manufacturer’s instructions.

### 4.8. Chemokine Analysis

Concentrations of pro-inflammatory chemokines in cell culture supernatant or peritoneal lavage of mice were analyzed using LEGENDplex kit (Mouse Proinflammatory Chemokine Panel, Biolegend, San Diego, CA, USA) according to the manufacturer’s instructions.

### 4.9. Quantitative Real-Time PCR

To quantify gene expression levels, isolated RNA was analyzed using quantitative real-time PCR as described previously [34]. Data were normalized to RPLP0-RNA expression. Relative expression levels of the target transcript were calculated compared to the reference transcript by using the ΔΔCT method [35]. All primers were purchased by Eurofins (Hamburg, Germany; Table 2).

### 4.10. RNA Sequencing

RNA of macrophages was isolated using RNeasy Micro Kit (Qiagen, Hilden, Germany) according to the manufacturer´s instructions. RNA quality was controlled using a RNA 6000 Pico chip (Agilent Technologies, Waldbronn, Germany) and RNA concentration was determined using Qubit HS RNA Assay Kit (Thermo Fisher Scientific, Waltham, MA, USA). Sequencing libraries were prepared using QuantSeq 3′ mRNA-Seq Library Prep Kit with the UMI Second Strand Synthesis Module for QuantSeq (Lexogen, Vienna, Austria). Quality of the cDNA was verified using Agilent DNA High Sensitivity DNA Chip (Agilent Technologies). DNA concentration was determined by Qubit dsDNA HS Assay Kit (Thermo Fisher Scientific). RNA sequencing was performed using a High Output Kit v2 on a NextSeq 500 sequencer (Illumina, San Diego, CA, USA). Bluebee Data Analysis Platform was used to analyze data. Data were mapped to mus musculus genome GRCm38 using Lexogen Quantseq 2.3.1. FWD UMI pipeline. Differential gene expression was analyzed with Lexogen QuantSeq DE 1.3.1 pipeline. Differentially regulated pathways were analyzed using gene ontology analysis (Gorilla [36,37]; base mean > 30, *p*-value < 0.01, logfold > 2) and gene set enrichment analysis (GSEA [38,39]). Heatmaps were generated using Morpheus matrix visualization and analysis software (https://software.broadinstitute.org/morpheus, 20 August 2020).

### 4.11. Proliferation of Macrophages

Peritoneal macrophages were seeded and co-cultured with NIH 3T3 FV-hCasp8 cells as described above and confluence was measured every 4 h for 48 h using 10× objective in IncuCyte imaging system (Essen BioScience, Ann Arbor, MI, USA). Cell confluence was calculated using IncuCyte 2020B software. 0.05 × 10^6^ IC-21 cells/well were stained with 5 μM eFluor 670 (Invitrogen, Carlsbad, CA, USA) according to the manufacturer´s instructions, seeded in a 24 well plate and co-cultured with NIH 3T3 FV-hCasp8 cells in a transwell system (6.5 mm transwell with 5 μm pore size; Corning). After indicated time points, fluorescence intensity of IC-21 cells was measured using flow cytometry (BD, FACSCanto II or LSPFortessa). Data were analyzed using FlowJo version 10.6.2.

### 4.12. Statistical Analysis

Statistical significance was assessed using ANOVA followed by Bonferroni post-tests or two-tailed unpaired t-test using GraphPad Prism 7 for Windows (GraphPad Software, La Jolla, CA, USA, * *p* < 0.05; ** *p* < 0.01; *** *p* < 0.001.

## Figures and Tables

**Figure 1 ijms-22-02230-f001:**
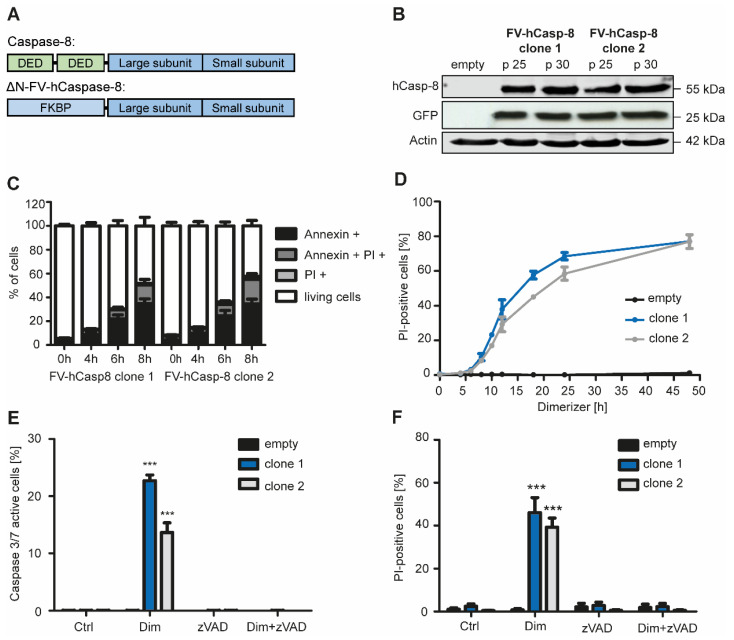
Generation and validation of an inducible cell death system of apoptosis. (**A**) DED domains of human caspase-8 were replaced by the dimerizer domain FKPB. Addition of dimerizer to NIH 3T3 cells expressing FV-hCaspase-8 leads to dimerization of caspase-8 and induction of apoptosis. (**B**) Two different single clones of NIH 3T3 cells expressing FV-hCaspase-8 were selected and protein expression of Caspase-8, GFP and Actin were analyzed by Western Blotting after 25 or 30 passages (p). (**C**) Annexin V and PI positive cells were analyzed over time using flow cytometry after treatment with 10 nM dimerizer (*n* = 4). (**D**) 10 nM dimerizer were added to NIH 3T3 cells and cell death was observed over time by counting PI-positive cells (*n* = 3). (**E**) Active Caspase 3 or 7 positive cells were counted 10 h after treatment with dimerizer and/or zVAD.FMK (*n* = 3). (**F**) The pan-caspase inhibitor zVAD.FMK was added and cells were treated for 18 h. PI-positive cells were counted (*n* = 3). Mean ± SEM are shown, *** *p* < 0.001.

**Figure 2 ijms-22-02230-f002:**
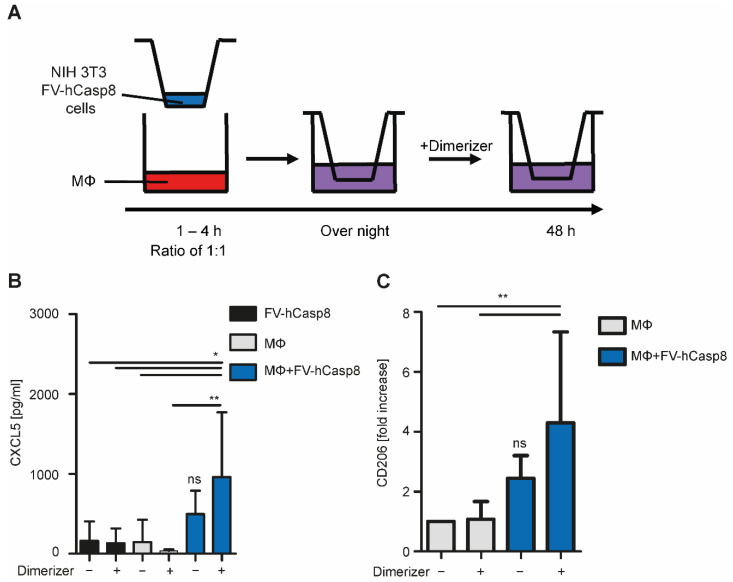
Establishing of a co-culture system. (**A**) Depiction of the experimental setup: Peritoneal macrophages (MΦ) were isolated from C57 BL/6 mice and co-cultured with NIH 3T3 cells at a ratio of 1:1 in a transwell system. Cells were co-cultured over night before treatment with 10 nM dimerizer. After 48 h, cytokines and chemokines in the supernatant and RNA of the macrophages were analyzed. (**B**) Concentration of CXCL5 was measured (*n* = 6); ns - not significant; * *p* < 0.05; ** *p* < 0.01. (**C**) mRNA levels of CD206 were quantified. Fold increase to control (MΦ) are shown (*n* = 8); ns - not significant; ** *p* < 0.01. Data are summarized from experiments using single clone 1 and 2. Mean ± SEM are shown.

**Figure 3 ijms-22-02230-f003:**
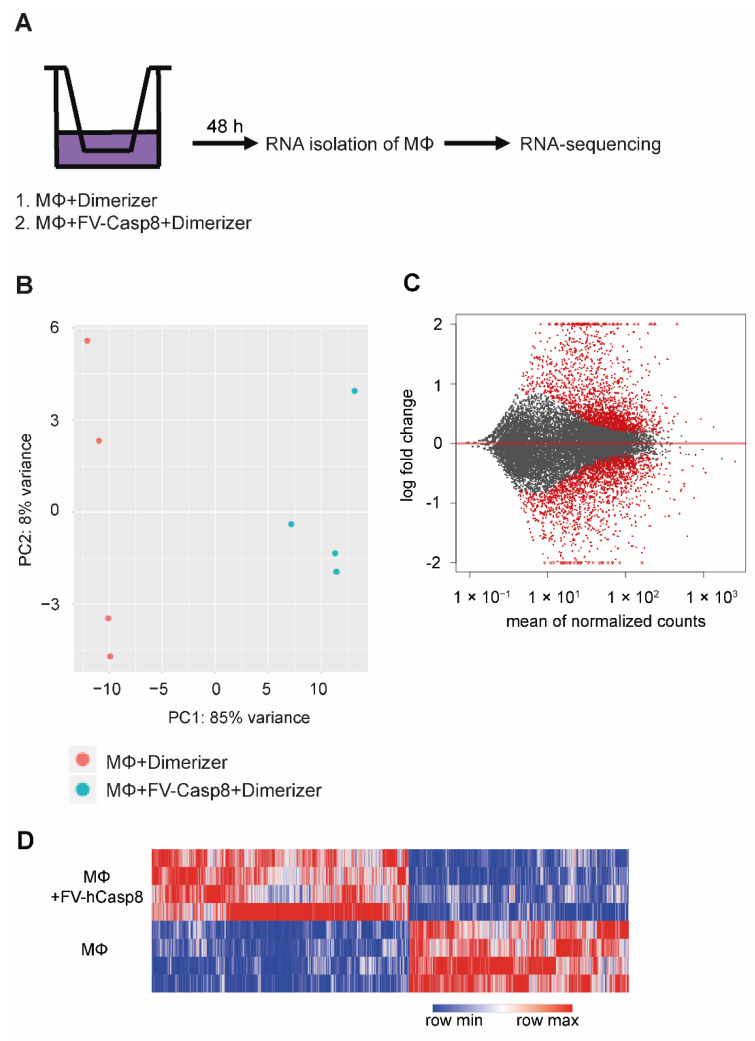
mRNA-sequencing of macrophages co-cultured with apoptotic cells. (**A**) Depiction of the experimental setup of RNA sequencing: Peritoneal macrophages (MΦ) were co-cultured with or without NIH 3T3 cells expressing FV-hCasp8 and incubated with dimerizer for 48 h in a transwell system. mRNA of macrophages was isolated and RNA sequencing was performed. (**B**) Principal component analysis (PCA) plot shows the variance of sample groups regarding their gene expression. (**C**) MA plot depicting the fold change of all genes against the mean of normalized counts of all samples is shown (grey dots). Red dots represent genes with a *p*-value smaller than 0.001, red triangles represent genes with log2 fold <−2 or >2. (**D**) Hierarchical clustering and heat map of differentially expressed genes are shown. 2905 genes were differentially expressed with a base mean higher than 30 and a *p* value lower than 0.05 (*n* = 4).

**Figure 4 ijms-22-02230-f004:**
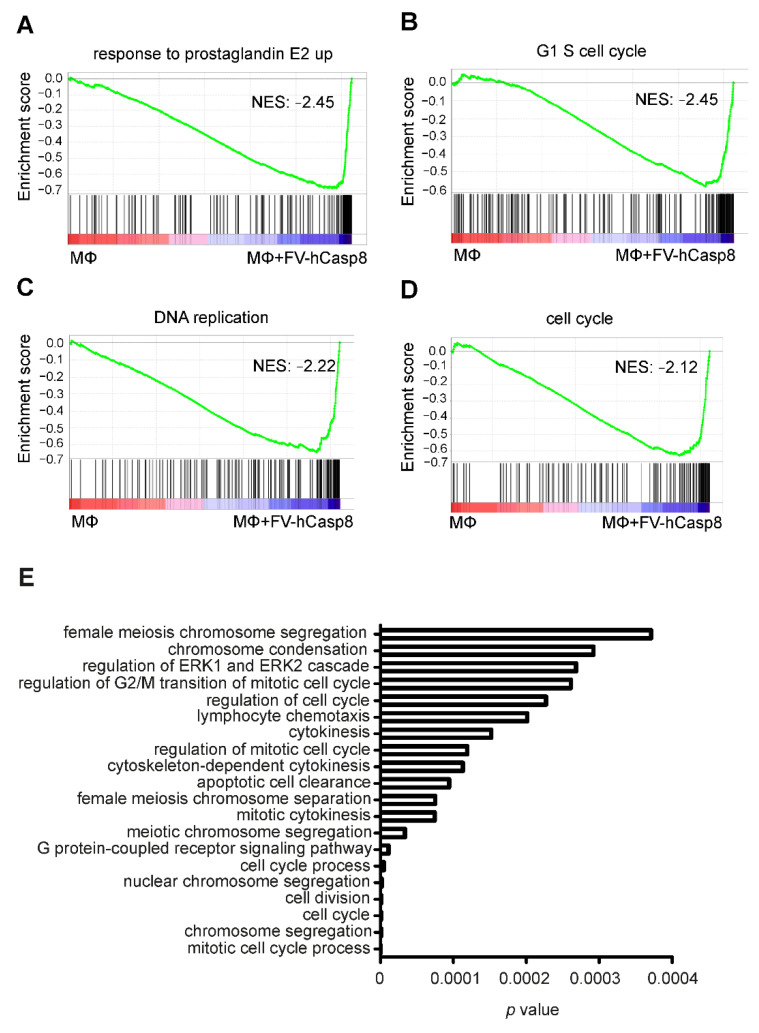
Macrophage pathways regulated by exposure to apoptotic cells. (**A**–**D**) Exemplary enrichment plots after gene set enrichment analysis are shown. GSEA shows enrichment of response to prostaglandin E2, G1 S cell cycle, DNA replication, and cell cycle pathways. Normalized enrichment scores (NES) can be found in the corresponding graph (*p* value < 0.01, FDR < 0.01). (**E**) Top 20 upregulated pathways in macrophages co-cultured with apoptotic cells after GO analysis using GO Gorilla are depicted.

**Figure 5 ijms-22-02230-f005:**
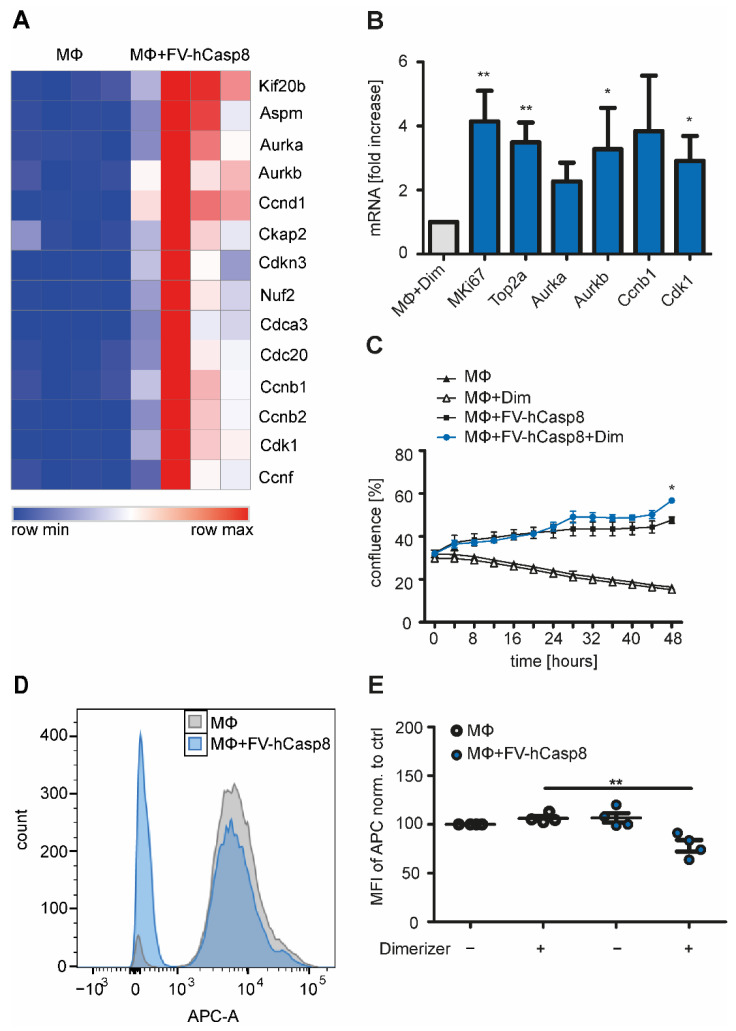
Peritoneal macrophages are proliferating in vitro after co-culture with apoptotic cells. (**A**) Heat map shows regulation of 14 genes associated with activation of cell cycle. (**B**) Macrophages (MΦ) were co-cultured with or without apoptotic cells and mRNA levels of cell cycle genes were quantified. Fold increase to control (MΦ+Dimerizer) normalized to RPLP0 is shown (*n* = 4, 5). (**C**) Confluence of peritoneal macrophages co-cultured with apoptotic cells was determined over 48 h (*n* = 3). (**D**) IC-21 cell were stained with eFluor 670 and co-cultured with apoptotic cells for 24 h. The histogram shows the eFluor 670 (APC) fluorescence of IC-21 cells co-cultured with living or apoptotic NIH 3T3 cells. (**E**) The mean fluorescence intensity of eFluor 670 (APC) of IC-21 cells co-cultured with living or apoptotic NIH 3T3 cells was measured (*n* = 4). Mean ± SEM are shown. * *p* < 0.05; ** *p* < 0.01.

**Figure 6 ijms-22-02230-f006:**
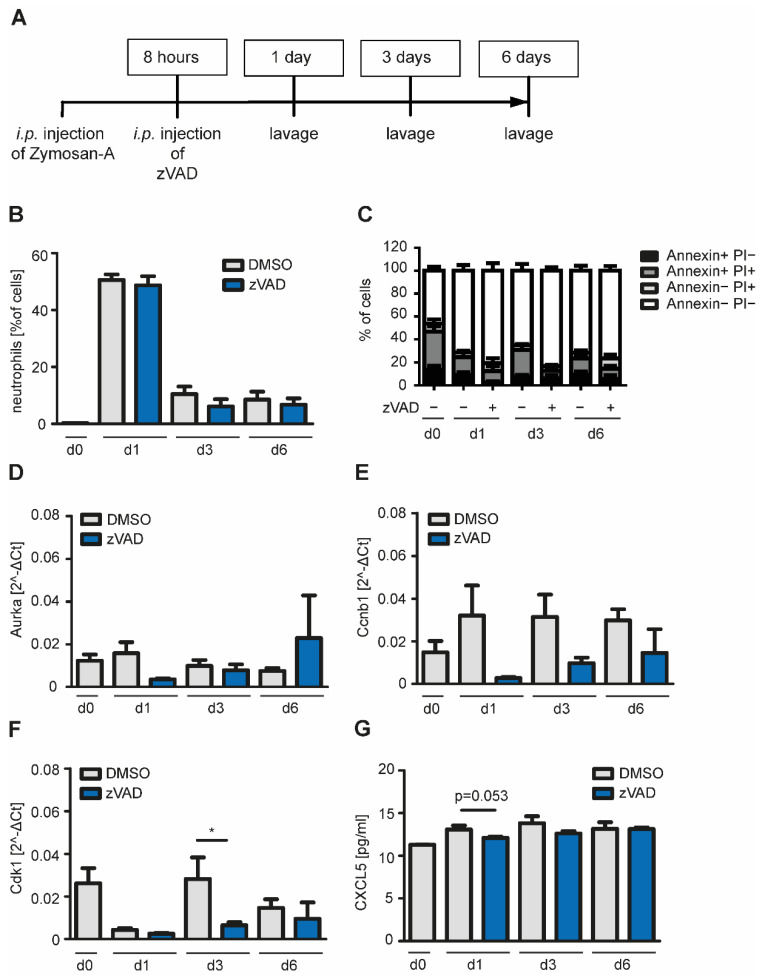
Inhibition of apoptosis in a model of Zymosan-induced peritonitis in mice. (**A**) Zymosan-A was injected *i.p.* into mice to induce self-resolving peritonitis. After 8 h, pan-Caspase inhibitor zVAD.FMK was injected *i.p.* to inhibit apoptosis. At day one, three or six after Zymosan-A injection cells of the peritoneum were harvested by lavage with 3 mL PBS. (**B**) Neutrophils were defined as CD45^+^Ly-6G^+^Ly-6C^+^MHC-II^-^ and percentage of neutrophils in the lavage was determined by flow cytometry (*n* = 3–9). (**C**) Cells of the lavage were stained with Annexin V/PI and analyzed using flow cytometry (*n* = 5–10). (**D**–**F**) Cells of the lavage were seeded and adherent macrophages were selected by washing with PBS. RNA of macrophages was isolated and levels of Aurka, Ccnb1 and Cdk1 were quantified using RT-PCR (*n* = 3–6). (**G**) Concentration of CXCL5 in the lavage was measured using Legendplex assay and flow cytometry (*n* = 4–9). Mean ± SEM are shown; * *p* < 0.05.

**Table 1 ijms-22-02230-t001:** To analyze immune cells in the peritoneum the following antibodies were used.

Marker	Dye	Company
CD16/32	none	BD
CD3	PE-CF594	BD
CD4	BV711	Biolegend
CD8	BV650	Biolegend
CD11b	BV605	BD
CD11c	AlexaFluor 700	BD
CD19	APC-H7	Biolegend
CD34	FITC	BD
CD45	VioBlue	Miltenyi
CD90.2	PE	Miltenyi
CD117	FITC	BD
CD140a	PE	BD
GITR	FITC	Biolegend
F4/80	PE-Cy7	Biolegend
HLA-DR (MHC II)	APC	Biolegend
Ly-6C	PerCP-Cy5.5	Biolegend
Ly-6G	APC-Cy7	Biolegend
NK1.1	AlexaFluor 700	BD

**Table 2 ijms-22-02230-t002:** Sequences of primers for RT-PCR.

Primer	Sequence
mCD206_fwd	CCATCTCAGTTCAGACGGCA
mCD206_rev	ACGGAAGCCCAGTCAGTTTT
mMki67_fwd	AGGAATCGCGGGAGACACAGCT
mMki67_rev	CCATTCCACCGCGCCATCTCTC
mTop2a_fwd	CGCTGGTTTTGTCGCTTTCCGG
mTop2a_rev	TACAGGCTGCAGCGGTGACAAC
mAurka_fwd	AGACCACTGTTCCCTTCGGTCC
mAurka_rev	CTGGCCACTGCTAGCAGATCCT
mAurkb_fwd	GTCTGGCCTGAACACGTTGTCC
mAurkb_rev	GGACTGGCTGTTGAACCGGTTC
mCdk1_fwd	CGAGGAAGAAGGAGTGCCCAGT
mCdk1_rev	AGCACATCCTGCAGGCTGACTA
mCcnb1_fwd	CTGAGCCTGAGCCTGAACCTGA
mCcnb1_rev	CCATCGGGCTTGGAGAGGGATT
mRPLP0_fwd	GCTGATCATCCAGCAGGTGT
mRPLP0_rev	GGACACCCTCCAGAAAGCGA

## Data Availability

The transcriptomic datasets generated during and/or analyzed during the current study are available at GEO: GSE164709.

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
