# Peer review of "Apoptotic Cells induce Proliferation of Peritoneal Macrophages"

_ijms, 2021, doi:10.3390/ijms22052230_

Round 1

Reviewer 1 Report

This study demonstrated the effect of apoptotic cells on macrophages using dimerization-induced cell model of caspase 8 in immune cell proliferation and inflammation. The study data are quiet interesting and the novelty of the research is considered to be very high.  I recomment to accept this manuscript for publication in this version.

Author Response

Thank you very much for your encouraging comments.

Reviewer 2 Report

I read with interest the Manuscript titled "Apoptotic cells induce proliferation of peritoneal macrophages". The topic is really interesting and attracts the reader's attention. The manuscript provides data about the direct and prominent impact of apoptotic cells on peritoneal macrophage proliferation. The authors established a co-culture system to study the effect of apoptotic cells on macrophages. The conception of the study is adequate, the introduction contains enough information to understand the background of the theme. The authors used several different techniques (cell sorting, cell culturing and treatment, flow cytometry, western blot analysis, quantitative real-time PCR, and RNA sequencing) to properly explore the topic. 

This manuscript is worth publication, however, the following minor points need to be discussed:

Following the 48 h co-culturing of the peritoneal macrophages with NIH 3T3 FV-h Casp8 cells the polarization marker CD206 mRNA level is elevated. Did the authors measure the CD206 receptor expression on the surface of macrophages in this condition?

Figure 3C is a little confusing. Maybe the resolution of the figure in my version of the reason, but it is really hard to recognize the difference between the dots and triangles. And what symbolize the grey dots?

The authors refer to the discussion that the tissue repair program in macrophages can be induced by IL-4. Did the authors measure the level of IL-4 from the lavage at day 3 or 6 after Zymosan-A injection?

In table 1. there is a lot of markers used to determination of the composition of immune cells, but the authors did not mention any information about these cells. Why is necessary to indicate these immune cell marker if no other data are available?   

How the authors isolated the peritoneal macrophages? Please add the sorting technique and markers of the separation of the macrophage population from peritoneal cells!

Author Response

Rebuttal letter

We are grateful for the constructive and encouraging comments of the referees. Addressing the queries or Reviewer 2 certainly improved our manuscript. Changes in the revised version of our manuscript have been marked in yellow and are substantiated in our point-by-point reply below.

Reviewer 2

Comment 1:

Following the 48 h co-culturing of the peritoneal macrophages with NIH 3T3 FV-h Casp8 cells the polarization marker CD206 mRNA level is elevated. Did the authors measure the CD206 receptor expression on the surface of macrophages in this condition?

Response:

This is an excellent suggestion. We certainly would have liked to conduct this experiment, but it was unfortunately not feasible due to technical issues with preparing (detaching without loss of viability) peritoneal macrophages for flow cytometry after co-culture with NIH 3T3 cells.

Comment 2:

Figure 3C is a little confusing. Maybe the resolution of the figure in my version of the reason, but it is really hard to recognize the difference between the dots and triangles. And what symbolize the grey dots?

Response:

We increased the resolution of Figure 3C and edited the description to allow data interpretation (please see Figure Legend to Figure 3).

Comment 3:

The authors refer to the discussion that the tissue repair program in macrophages can be induced by IL-4. Did the authors measure the level of IL-4 from the lavage at day 3 or 6 after Zymosan-A injection?

Response:

Again, the reviewer raises an important point. Unfortunately levels of IL-4 in the peritoneal lavage of mice in our hands were below detection limits.

Comment 4:

In table 1. there is a lot of markers used to determination of the composition of immune cells, but the authors did not mention any information about these cells. Why is necessary to indicate these immune cell marker if no other data are available?

Response:

We agree with the reviewers and now show data for all immune cell subsets measured in new Figure S2 and Figure S3 (mentioned on p. 9).

Comment 5:

How the authors isolated the peritoneal macrophages? Please add the sorting technique and markers of the separation of the macrophage population from peritoneal cells!

Response:

We now specify the protocol to clearly indicate how the isolation of peritoneal macrophages was performed on p. 13.
